# Addressing the Length Bias Problem in Document-Level Neural Machine Translation

**Zhuocheng Zhang[1,2][†], Shuhao Gu[1,2][†], Min Zhang[3], Yang Feng[1,2][∗]**
[1]Key Laboratory of Intelligent Information Processing,
Institute of Computing Technology, Chinese Academy of Sciences (ICT/CAS)
[2]University of Chinese Academy of Sciences, China
[3]School of Future Science and Engineering, Soochow University, China
zhangzhuocheng20z@ict.ac.cn
shuhaog515@gmail.com
zhangminmt@hotmail.com
fengyang@ict.ac.cn

## Abstract

Document-level neural machine translation (DNMT) has shown promising results by incorporating more context information. However, this approach also introduces a length bias problem, whereby DNMT suffers from significant translation quality degradation when decoding documents that are much shorter or longer than the maximum sequence length during training. To solve the length bias problem, we propose to improve the DNMT model in training method, attention mechanism, and decoding strategy. Firstly, we propose to sample the training data dynamically to ensure a more uniform distribution across different sequence lengths. Then, we introduce a length-normalized attention mechanism to aid the model in focusing on target information, mitigating the issue of attention divergence when processing longer sequences. Lastly, we propose a sliding window strategy during decoding that integrates as much context information as possible without exceeding the maximum sequence length. The experimental results indicate that our method can bring significant improvements on several open datasets, and further analysis shows that our method can significantly alleviate the length bias problem[1].

## 1 Introduction

Document-level neural machine translation (DNMT) (Gong et al., 2011; Hardmeier et al., 2013; Garcia et al., 2015; Miculicich et al., 2018; Tan et al., 2019; Maruf et al., 2019; Zheng et al., 2020; Xu et al., 2020) is proposed to enhance translation quality by leveraging more contextual information. Recently, the document-to-document (doc2doc) DNMT model (Junczys-Dowmunt,

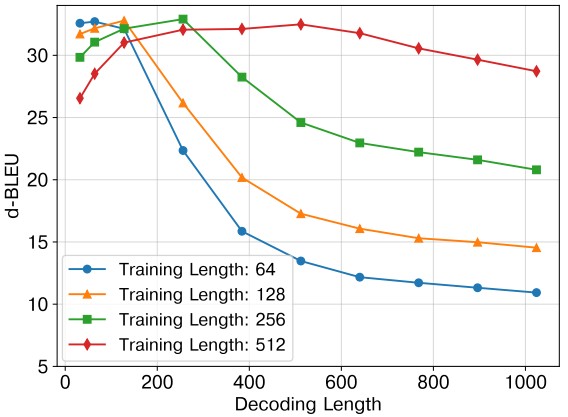

Figure 1: The length bias problem for doc2doc DNMT model. The translation quality degrades significantly as the decoding length deviates from the training length.

2019; Liu et al., 2020; Bao et al., 2021; Sun et al., 2022b), which expands the translation scope from individual sentences to entire documents, has demonstrated exceptional performance, thereby drawing increased attention. For the training of doc2doc DNMT model, multiple sentences are assembled into sequences that are close to the predetermined maximum length, enabling the model to learn information from the context as much as possible. However, this training strategy can lead to overfitting to the maximum length. Sequences that are significantly shorter than the maximum length may be overlooked by the model due to their smaller proportion in the training set. Besides, the model also lacks the ability to handle the sequences that exceed the maximum length, which are not encountered by the model during training. Consequently, the length bias problem results in a significant degradation in translation quality when the length of the decoded sequence deviates from the maximum sequence length, which is shown in Figure 1.

---

[∗]Corresponding author.
[†]Equal contribution.
[1]Code is at https://github.com/ictnlp/LengthBiasDNMT.

Some researchers have made their attempts to enhance the length generalization capabilities of DNMT model from various perspectives. Some approaches employ data augmentation techniques (Junczys-Dowmunt, 2019; Sun et al., 2022b) to mix documents with shorter segments such as sentences or paragraphs, thereby augmenting the diversity of sequence lengths in the training set. However, the proposed augmentation method does not necessarily guarantee a balanced length distribution, as the length distribution is still influenced by the training corpus itself. Bao et al. (2021) incorporates a locality assumption as an inductive bias into the Transformer model, which reduces the complexity of target-to-source attention. As a result, their method allows for the setting of larger maximum lengths, thereby augmenting the model's ability to handle longer documents. However, this method can only bring limited improvements for the short sequences. Besides, the aforementioned approaches are still incapable of directly handling sequences that exceed the maximum sequence length during testing and still require segmentation of excessively long test sequences.

Given above, we aim to enhance the capability of our model to handle both long and short sequences. Additionally, we seek to enable the model to directly translate sequences that exceed the maximum length, thereby avoiding information loss caused by segment truncation. To achieve these objectives, we have made improvements in the sampling of training data, attention weight computation, and decoding strategies. During training, we first sample the sequence lengths and then construct the training data accordingly. This dynamic variation in sequence lengths within different epochs ensures that the model encounters a more balanced distribution of sequence lengths during training. Furthermore, we introduce a scaling factor during attention computation to ensure that, even as the sequence length increases, the model can still focus on relevant target information and prevent attention divergence. Lastly, when decoding sequences that exceed the maximum length, we employ a sliding window decoding strategy which allows for the retention of more context information while ensuring that the context length remains below the maximum sequence length. These three proposed methods collectively contribute to improving the length generalization capabilities of the DNMT model from different perspectives. Moreover, their

combined application yields further performance enhancements. We conduct experiments on several document-level open datasets and the experimental results indicate that our method can bring significant improvements. Further analysis shows that our method can significantly alleviate the length bias problem.

## 2 Background

In this section, we will give a brief introduction to the Transformer (Vaswani et al., 2017) model and the doc2doc DNMT model.

### 2.1 The Transformer

The transformer model is based on the encoder-decoder architecture. The encoder is composed of $N$ identical layers. Each layer has two sublayers. The first is a multi-head self-attention sublayer, and the second is a fully connected feed-forward network. Both of the sublayers are followed by a residual connection operation and a layer normalization operation. The decoder is also composed of $N$ identical layers. In addition to the same kind of two sublayers in each encoder layer, the cross-attention sublayer is inserted between them, which performs multi-head attention over the output of the encoder.

The attention mechanism is the core part of the Transformer model, which is computed as:

$$\text{Attention}(\mathbf{Q}, \mathbf{K}, \mathbf{V}) = \text{softmax}\left(\frac{\mathbf{Q}\mathbf{K}^\top}{\sqrt{d_k}}\right)\mathbf{V},$$
(1)

where $\mathbf{Q}, \mathbf{K}$ and $\mathbf{V}$ represent the query, key, and value vectors, respectively. $d_k$ denotes the dimension of the key vectors. The softmax function is applied to normalize the dot-product similarities between the queries and keys, and the result is multiplied by the value vectors to obtain the weighted sum.

### 2.2 The doc2doc DNMT model

The doc2doc DNMT model is to translate the whole document directly. Different from the conventional DNMT model, which translates documents sentence by sentence with an additional context encoder (Tan et al., 2019; Maruf et al., 2019; Yang et al., 2019; Zheng et al., 2020; Xu et al., 2020; Yun et al., 2020), multiple sentences will be simultaneously input into the doc2doc DNMT model for training and decoding. The training data consists of different documents $\mathcal{D} = \bigcup_{i=1}^{n}\{\mathbf{d}_i\}$, where $n$

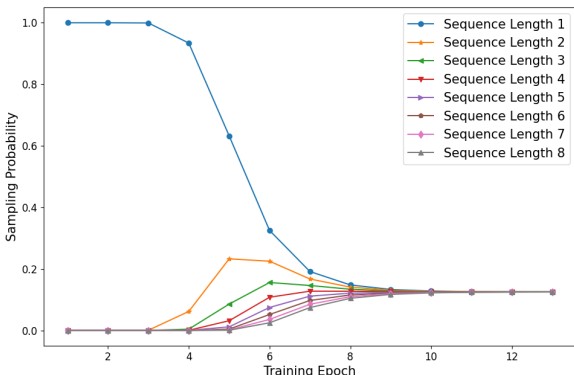

Figure 2: An example of the sampling probabilities of different sequence lengths during training.

denotes the number of documents in the training data. Each document $\mathbf{d}_i$ contains source and target sentences $\mathbf{d}_i = \bigcup_{j=1}^m \{(\mathbf{x}_{ij}, \mathbf{y}_{ij})\}$, where $m$ denotes the number of sentences in each document. Besides, special symbols are often inserted within the documents to distinguish between different sentences. The training objective can be written as:

$$\arg\max_{\theta} \sum_{i=1}^{n} \sum_{j=1}^{m} \sum_{k=1}^{|\mathbf{y}_{ij}|} P(y_{ij}^k | y_{ij}^{<k}, \mathbf{x}_i, \mathbf{y}_{i,<j}), \quad (2)$$

where $|\mathbf{y}_{ij}|$ denotes the number of the words in the $j$-th target sentence of the $i$-th document. During decoding, documents that exceed the maximum sequence length are also segmented, otherwise it will result in a significant decrease in translation quality.

## 3 Method

Our method aims to enhance the length generalization capabilities of the doc2doc DNMT model, thereby alleviating the length bias problem. To achieve this goal, we have made improvements in three aspects: training data sampling, attention computation, and decoding strategies.

### 3.1 Dynamic Length Sampling

Dynamic length sampling (DLS) aims to ensure that the model has the opportunity to encounter training sequences of various lengths throughout the training process, thereby facilitating better learning and retention of the ability to translate sequences of different lengths. Therefore, the key challenge of this method lies in determining the sampling probabilities for different sequence lengths. Given that translating complete documents usually involves longer input and output

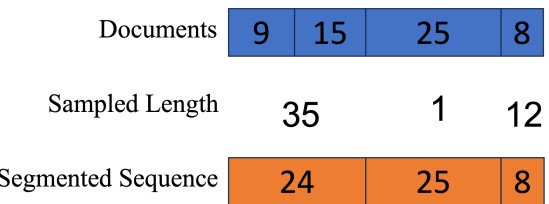

Figure 3: An example of the segmented sequences.

sequences, directly learning document-level translation is more difficult. Hence, in the initial stages of training, we focus more on training the model on shorter sequences, which improves training stability and accelerates model convergence. As training progresses, we hope to increase the probability of sampling longer sequences or documents, allowing the model to gradually learn longer contextual dependencies.

Following the above intuitions, we define the sampling probabilities of different lengths as:

$$p_l = \frac{w_l^{\frac{1}{T}}}{\sum_{l=1}^{L} w_l^{\frac{1}{T}}}, \quad (3)$$

where $L$ denotes the maximum sequence length and $w_l$ denotes the sampling weight assigned for different lengths which is defined as $w_l = e^{-l}$. $T$ is a sampling temperature (Arivazhagan et al., 2019), which is computed as $T = e^{(ep-\gamma)}$, where $ep$ denotes the current epoch number and $\gamma$ is a hyperparameter, which should be adjusted according to the dataset. The temperature $T$ varies with the training epoch. An example of the sampling probabilities of different sequence lengths during training is shown as in Figure 2, where $\gamma$ is set as $5$ and max length is set as $8$. We can see from the figure that in the initial stage of training, the probability of short sequence length being sampled is relatively high. In the later stages of training, the probability of different sequence lengths being sampled tends to be equal. Although in real training processes, the maximum sequence length is typically much greater than 8, the pattern of the sampling probabilities follows a similar trend.

Specifically, before the training of each epoch begins, we first update the probability of each sequence length being sampled according to Equation 3. Then, for each document $\mathbf{d}_i$ in the training set, we sample different sequence lengths $[l_{i1}, l_{i2}, \ldots, l_{ik}, \ldots]$. We segment the documents $\mathbf{d}_i$ into different sequences $\mathbf{s}_{ik}$ from left to right

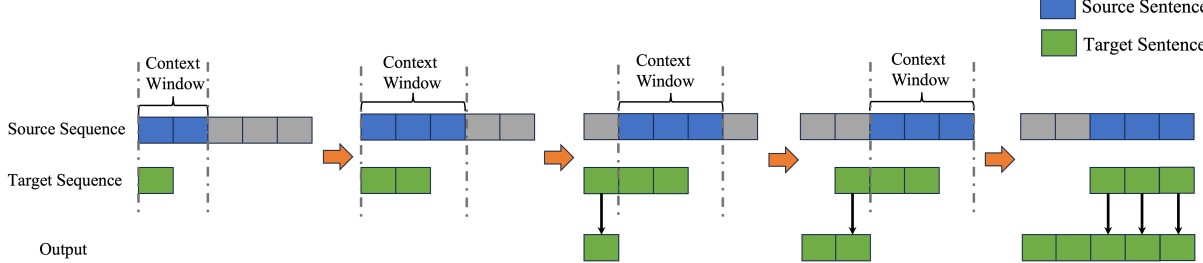

Figure 4: An illustration of the sliding decoding strategy. We set the window size as three sentences in this example for display convenience.

if the segmented length is shorter than the sampled length. But if the sampled sequence length is less than the current sentence length of the document, we will select the current single sentence as the input sequence. This overall process can be demonstrated as:

$$\mathbf{s}_{ik} = \{\mathbf{x}_{i,a:b}, \mathbf{y}_{i,a:b}\},$$
$$s.t. \begin{cases} |\mathbf{x}_{i,a:b}| \le l_{ik}, |\mathbf{y}_{i,a:b}| \le l_{ik}, & a < b, \\ \quad\quad or \\ |\mathbf{x}_{i,a:b}| > l_{ik}, |\mathbf{y}_{i,a:b}| > l_{ik}, & a = b \end{cases} \quad (4)$$

Different sequences $\mathbf{s}_{ik}$ don't overlap with each other. We show an example of the above process in Figure 3. In this example, the document has 4 sentences, with lengths of 9, 15, 25, and 8, respectively. The sampled lengths are 35, 1, and 12, respectively. Therefore, the final input sequence we obtained contains three segments, with lengths of 24, 25, and 8, respectively. After processing the entire training set, we can obtain the sequences required for the current epoch.

### 3.2 Length Aware Attention

The role of the attention mechanism is to retrieve information from the target sequence that is relevant to itself (self-attention) or the current translation (cross-attention). However, the DNMT model usually needs to handle a wide range of context except for the target sequence, which may interfere with the normal operation of the attention mechanism, leading to the divergence of the attention results and consequently deteriorated translation quality for long segments.

Inpired by Chiang and Cholak (2022), we propose the length aware attention (LAA), which adds a scaling factor to the original attention computation (Equation 1) to mitigate this issue:

$$\text{Attention} = \text{softmax}\left(\frac{\mathbf{Q}\mathbf{K}^\top}{\sqrt{d_k}} * \underset{\iota}{log\, l}\right)\mathbf{V}, \quad (5)$$

where $l$ denotes the length of the attended sequence and $\iota$ denotes the average length of sequences in the current training epoch. Because the sequence lengths are sampled per epoch by DLS, $\iota$ also changes gradually. It can be demonstrated that incorporating the aforementioned length scale effectively mitigates the issue of entropy divergence in attention results when dealing sequences with different lengths. We have included the proof process in A. During decoding, $\iota$ is set as the value corresponding to the final epoch of the training phase.

### 3.3 Sliding Decoding

During training, the DNMT model needs to set a maximum sequence length. However, during the decoding phase, it often encounters documents that exceed this maximum length. Directly decoding such long documents can lead to inferior results, as the model has not been exposed to documents exceeding the maximum length during training. A common approach is to split the long document into shorter segments and translate them separately, subsequently concatenating the translation results. However, such segmentation may result in the loss of contextual information, thereby affecting translation quality.

To address these issues, we propose a method that utilizes a sliding window for decoding (SD). Specifically, when the length of the input sequence is smaller than the maximum length, the complete sequence, including the target sentence and the context, is used for translation. However, when the input sequence exceeds the maximum length, we discard the oldest source-side context information from the current time step onwards and no longer employ it to assist in translation. Simultaneously, the corresponding oldest target-side context information is also discarded, but it will be preserved as part of the translation result. The illustration

of the overall process is shown in Figure 4. If we employ a beam search strategy during the decoding process, we retain the candidate with the highest generation probability within the current beam for output and subsequent decoding.

## 4 Experiments

### 4.1 Data Preparation

We conduct experiments on 3 most commonly used English to German (En→De) translation datasets. The description of the datasets are as follows:

- **TED** is provided by *IWSLT2017* (Cettolo et al., 2012), containing talks from TED. We adopt tst2016-2017 as the test sets, and the rest for the valid sets.

- **News** contains parallel documents extracted from *NewsCommentary* in news domain[2]. In our experiments, *newstest2015* and *newstest2016* are used for validation and test, respectively.

- **Europarl** is extracted from *Europarl v7* (Koehn, 2005) and split using *SPEAKER tags*. We follow the train/develop/test sets spliting as Maruf et al. (2019).

We use the Moses toolkit (Koehn et al., 2007) to tokenize other languages. Besides, integrating operations of 32K is performed to learn BPE (Sennrich et al., 2016). Following Bao et al. (2021); Liu et al. (2020), we set the maximum sequence length as 512 in our main experiments.

### 4.2 Systems

The systems used for comparision in our experiments are as follows:

- **Transformer** (Vaswani et al., 2017): We have obtained three systems with different training methods based on the Transformer model. The **Trans-sent** model is trained with the sentence-level corpus. The **Trans-doc** model is trained with the document-level training corpus. The **Trans-FT** model is fine-tuned based on the Transformer-sent model with the document-level corpus.

- **HAN** (Tan et al., 2019): This method employs a hierarchical attention mechanism in Transformer to capture contextual information at sentence-level and word-level.

- **Flat** (Ma et al., 2020): This methods feeds the concatenated sentences into a pre-trained BERT to collect the contextualized representations of the sentence being translated.

- **LED** (Beltagy et al., 2020): The proposed model in this method is equipped with a well designed sparse attention mechanism. We reproduce this by using *transformers*[3].

- **Doc-Trans** (Zhang et al., 2018): This method introduces a new context encoder to represent document-level context. They also propose a two-step training approach to effectively utilize abundant sentence-level parallel corpora.

- **G-Transformer** (Bao et al., 2021): This method incorporates a locality assumption as an inductive bias into the Transformer model. We train the model with the document-level corpus from scratch (**G-Trans**) and also pretrain the model with sentence-level corpus and then fine tune the model with the document-level corpus (**G-Trans-FT**).

- **MR** (Sun et al., 2022b): This method splits each document averagely into different parts for multiple times and collect all the sequences for training.

- **ALiBi** (Press et al., 2021): This method improves length generalization by adding static non-learned bias to attention weights. We train the model with the document-level corpus from scratch (**ALiBi**) and also pretrain the model with sentence-level corpus and then fine tune the model with the document-level corpus (**ALiBi-FT**).

- **Our System**: We applied the proposed methods, including dynamic length sampling (DSL), length aware attention (LAA) and sliding decoding (SD), to the Transformer-doc model (**Trans-doc+Ours**) and the G-Transformer model (**G-Trans+Ours**), respectively.

---

[2]https://www.casmacat.eu/corpus/news-commentary.html

[3]https://github.com/huggingface/transformers

| Models | TED | | | | Europarl | | | | News | | | |
|---|---|---|---|---|---|---|---|---|---|---|---|---|
| | s-BLEU | d-BLEU | s-chrF | d-chrF | s-BLEU | d-BLEU | s-chrF | d-chrF | s-BLEU | d-BLEU | s-chrF | d-chrF |
| Trans-sent | 24.12 | 28.02 | | | 30.33 | 32.45 | | | 24.91 | 26.94 | | |
| Trans-doc | 18.51 | 25.20 | | | 30.86 | 33.11 | | | 21.11 | 24.02 | | |
| HAN | 23.79 | 28.17 | | | 30.74 | 32.90 | | | 24.22 | 26.31 | | |
| Flat | 24.32 | 28.17 | | | 30.92 | 33.04 | | | 24.85 | 26.88 | | |
| LED | 18.46 | 24.29 | | | 29.90 | 32.48 | | | 12.13 | 16.63 | | |
| Doc-Trans | 23.81 | 27.64 | | | 30.74 | 32.88 | | | 24.79 | 26.77 | | |
| G-Trans | 22.53 | 25.90 | | | 32.02 | 34.14 | | | 23.87 | 25.90 | | |
| MR | 23.99 | 28.61 | 53.73 | 70.54 | 31.54 | 33.75 | 61.36 | 69.83 | 24.79 | 27.14 | 54.06 | 64.33 |
| ALiBi | 19.65 | 26.30 | 47.22 | 68.26 | 29.99 | 32.54 | 60.19 | 69.28 | 12.67 | 22.83 | 36.19 | 59.91 |
| ALiBi-FT | 20.85 | 27.55 | 49.54 | 69.71 | 29.64 | 32.55 | 59.76 | 69.41 | 17.37 | 24.02 | 43.49 | 60.79 |
| Trans-FT | 24.31 | 28.48 | 54.70 | 70.51 | 31.16 | 33.58 | 61.29 | 69.91 | 23.96 | 27.33 | 53.27 | 64.98 |
| Trans-doc + DLS + LAA | 24.93 | 28.95 | 55.18 | 70.69 | 31.85 | 34.32 | 61.26 | 70.01 | 24.57 | 28.52 | 53.13 | **65.47** |
| Trans-doc + Our method | 24.60 | 28.43 | 55.16 | 70.55 | 31.63 | 34.33 | 60.91 | 69.93 | 23.72 | 27.95 | 52.50 | 65.20 |
| G-Trans-FT | 25.07 | 28.86 | 55.65 | 70.79 | 32.38 | 34.51 | 62.15 | **70.32** | 25.87 | 27.82 | 55.71 | 65.14 |
| G-Trans + DLS + LAA | **25.37** | **29.07** | **55.76** | **70.81** | 32.67 | 34.81 | **62.06** | 70.23 | **26.70** | **28.66** | **55.96** | 65.24 |
| G-Trans + Our method | 24.87 | 28.53 | 55.34 | 70.51 | **32.67** | **34.82** | 61.99 | 70.17 | 26.56 | 28.52 | 55.78 | 65.09 |

Table 1: The experimental results of our proposed method on *TED*, *Europarl* and *News*. The best score are shown in **bold**. For our proposed Trans-doc + Our and G-Trans + Our, the documents are translated as a full unit without segmentation, while for other methods, the documents are segmented according to the maximum sequence length of 512.

| | Components | | | Scores | | | |
|---|---|---|---|---|---|---|---|
| ID | DLS | LAA | SD | s-BLEU | d-BLEU | s-chrF | d-chrF |
| 1 | ✔ | ✔ | ✔ | 31.63 | 34.33 | 60.91 | 69.93 |
| 2 | ✔ | ✔ | ✘ | 31.85 | 34.32 | 61.26 | 70.01 |
| 3 | ★ | ✘ | ✘ | 31.86 | 34.21 | 61.33 | 69.84 |
| 4 | ✔ | ✘ | ✘ | 32.06 | 34.37 | 61.52 | 69.94 |
| 5 | ✘ | ✔ | ✘ | 31.36 | 33.72 | 61.26 | 69.93 |
| 6 | ✘ | ✘ | ✘ | 31.16 | 33.58 | 61.29 | 69.91 |

Table 2: The ablation study of our proposed method. We conduct ablation study on *Europarl* with Trans-FT. The marker ✔ and ✘ indicate the component is involved and not involved, respectively. The marker ★ indicate the length sampling is applied without dynamic adjusting the temperature.

**Implementation Details** All the systems are implemented as the base model configuration in Vaswani et al. (2017). We train our system on 4 NVIDIA 3090 GPUs by using Adam (Kingma and Ba, 2017) optimizer. Most training parameters are kept the same with Bao et al. (2021), where the learning rate $lr = 5e - 4$, $\beta_1 = 0.9$, $\beta_2 = 0.98$. The warmup step is set to 4000 and the label smoothing (Szegedy et al., 2015) value is set to 0.1. The dropout ratio is set to 0.3 on *TED* and *News*, and 0.1 on *Europarl* for its larger scale. During decoding, we set the context window to 0.8 of the maximum sequence length used during the training phase to prevent performance degradation caused by overly long target sequences.

### 4.3 Main Results

During decoding, the test documents with a length less than the maximum length will be directly input into the model. Documents with a length greater than the maximum length will be segmented into several shorter sequences according to the maximum length and input into the model separately (Liu et al., 2020). We generate the translations with a beam size of 5 and length penalty $\alpha = 1$. We use the SacreBLEU tool (Post, 2018) to evaluate the output with s-BLEU (sentence BLEU) (Papineni et al., 2002), d-BLEU (document BLEU) (Liu et al., 2020), s-chrF (sentence-chrF) (Popović, 2015) and d-chrF (document-chrF), respectively. To make our experimental results comparable with previous studies (Sun et al., 2022b), our BLEU scores are calculated in a case-insensitive manner.

The main results are shown in Table 1. In the En→De translation task, our method outperforms the majority of the comparative systems, when in combination with the conventional doc2doc DNMT (Trans-doc+DLS+LAA). Furthermore, our proposed Trans-doc+DLS+LAA achieves performance comparable to the best-performing comparative system G-Trans-FT and surpass MR by a large margin. Further improvements are observed when integrating our method with G-transformer (G-Trans+DLS+LAA), and it can achieve state-of-the-art performance on all datasets. However, when combined with the slide decoding strategy, the performance drops slightly. We suspect that

this may be due to the error accumulation problem. On the other hand, the slide decoding strategy aims to solve the length extrapolation problem, and we further explore its advantages in Section 5.2.

## 5 Discussion

### 5.1 Ablation Study

To further understand the impact of each step of our method, we perform further studies by removing certain parts of our method. The results are given in Table 2. Upon comparing the performance of systems 2 and 6, it is evident that removing dynamic length sampling (DLS) significantly deteriorates the model's performance. This observation validates the importance of DLS and LAA in enhancing the system's performance. Furthermore, comparing the results of system 5 and 6, Length Aware Attention (LAA) also demonstrates a notable improvement, indicating that our method can effectively capture the contextual information. Lastly, when comparing system 3 and system 4, we find that dynamic adjust the temperature can further improve the performance without the need for fine-tuning. Therefore, the above results provide evidence that all of our methods are effective in enhancing the performance of doc2doc DNMT.

In addition, to verify the effectiveness of our proposed Length Aware Attention (LAA), we conducted a statistical analysis of the average entropy of the attention mechanism when translating sentences and documents of length 512. As shown in 4, it can be observed that the entropy of the attention mechanism is more stable after applying the LAA method, which indicates that after applying the LAA mechanism, the model demonstrates better consistency in handling sentence-level and document-level text. Furthermore, by applying the DLS and LAA method, the entropy of attention when translating the document is lower than that of the FT method, indicating that the model concentrates more on the long-range contexts.

### 5.2 Length Generalization

The main motivation of our approach is to enhance the length generalization performance of the doc2doc DNMT model, thereby addressing the issue of length bias. To assess the effectiveness of our method in achieving this goal, we conduct further analysis based on the English-German datasets. We decode the test set using the systems in the main experiments with different maximum lengths

and measured their corresponding d-BLEU scores. The results are presented in Figure 5. The results indicate that the baseline system and many comparison systems experience a significant decrease in d-BLEU score when the decoding length deviates from the maximum length used during training (512). In contrast, our method exhibits no significant decrease in BLEU score. This demonstrates that our approach can enhance the length generalization performance of the doc2doc model and alleviate the issue of length bias. In particular, when the decoding length exceeds the training length, the performance of the existing methods suffers from a huge drop, while our proposed slide decoding is able to maintain the high translation quality.

To further comprehend why our approach can enhance the length generalization performance of the model, we perform a visualization of the length distribution of the training data. We visualized the length distribution of the original corpus used for training, the data employed by the MR method, and the data used in the final epoch after incorporating DLS. The results are presented in Figure 6, which demonstrates that our approach achieves a more uniform length distribution. Consequently, our method has the capacity to improve the length generalization performance of the model to a greater extent.

### 5.3 The Discourse Phenomena

To investigate the translation of discourse phenomena, we conduct experiments on ContraPro test suite (Müller et al., 2018), a large contrastive test suite extracted from OpenSubtitles 2018 (Lison and Tiedemann, 2016)., to measure the translation accuracy of English pronoun "it" into the corresponding German translations "er", "sie" or "es". We employ *Europarl* as the training set, and the maximum sequence length is setting to 512. As shown Table 3, compared to random selection, the sentence-level translation model has the ability to infer a portion of the correct answer based on the information within the sentence. However, with the help of contextual information, document-level neural machine translation models outperforms sentence-level baseline by a large margin. Utilizing our proposed DLS and LAA, the error rate of Transformer-doc is further reduced, indicating that our approach can further enhance the capability of the model to capture the contextual information.

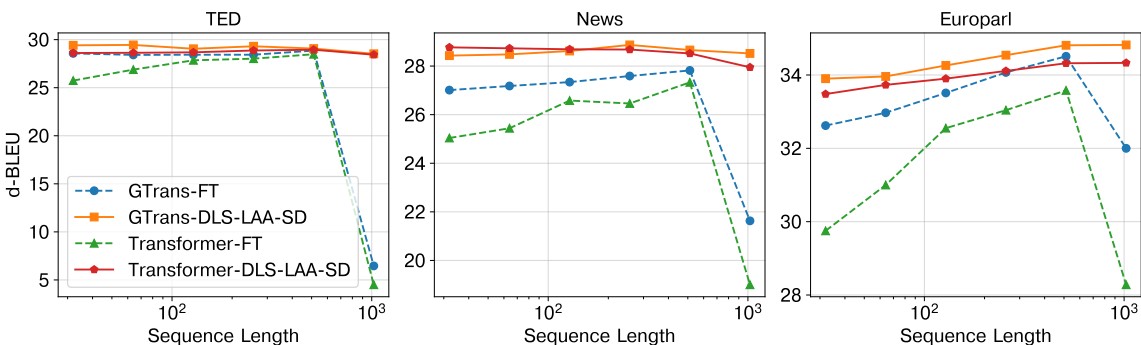

Figure 5: The length generalization of the different methods. We represent our method using solid lines while the baseline mathod using dashed lines.

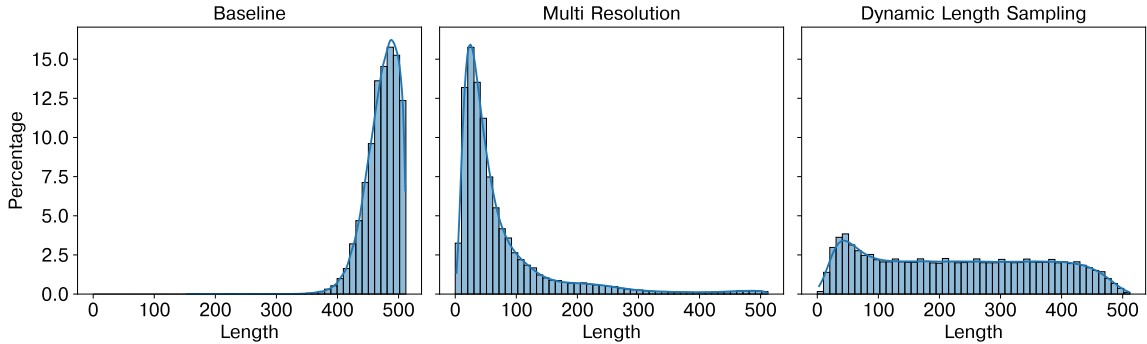

Figure 6: The length distributions of the training data used by different training strategies. We collect these distributions at the maximum length equal to 512.

| Method | ContraPro ACC(%) |
|---|---|
| Random | 33.33 |
| Trans-sent | 52.00 |
| Trans-FT | 70.58 |
| Trans-doc+DLS+LAA | 72.28 |

Table 3: The results of ContraPro test suit, measured by accuracy.

| Method | sentence | document | $\delta$ |
|---|---|---|---|
| Trans-FT | 2.65 | 4.0 | 1.35 |
| Trans-DLS | 2.51 | 3.95 | 1.44 |
| Trans-DLS-LAA | 2.74 | 3.94 | 1.20 |

Table 4: The average entropy of the attention mechanism when translating at sentence and document level. $\delta$ represents the difference in entropy when translating at different granularity.

## 6 Related Work

### 6.1 Document-Level Neural Machine Translation

Document-level neural machine translation can be broadly divide into two categories, including sentence-to-sentence (sen2sen) approach and document-to-document (doc2doc) approach (Maruf et al., 2021). The former feed the context as additional information to assist the translation of each single sentence in the document independently, which is also known as multi encoder method (Lupo et al., 2022). Jean et al. (2017) leveraged additional attention to capture the previous context; Kuang and Xiong (2018) proposed to control the usage of context by a gate function; Wang et al. (2017), (Miculicich et al., 2018), and (Zheng et al., 2020) introduced hierarchical attention networks to model the contextual information from the documents; Maruf et al. (2019) and Martins and Astudillo (2016) designed a selective attention network to extract most useful information from the massive context; Yang et al. (2019) proposed a query-guided capsule network to further

model the relationship between the context words. However, the scarcity of the datasets (Chen et al., 2021) and the sparsity of the contextual information make these model hard to be trained. Lupo et al. (2022) further address this problem by splitting the sentence into smaller pieces to augment the document-level corpus.

Another type of methods fall into doc2doc paradigm, which treats the entire document as a whole unit. Tiedemann and Scherrer (2017) proposed that by extending the translation granularity from sentence to documents the translation become more coherent; Liu et al. (2020) and (Ma et al., 2020) found that the translation quality could be improved by a large margin through incorporating pretraining; Bao et al. (2021) suggest that direct training a doc2doc transformer may fail to converge on small datasets, and proposed to solve this problem by incorporating group attention masks. Similarly, (Sun et al., 2022b) proposed to tackle the same problem by expanding the dataset with a multi-resolution (MR) strategy. On the other hand, this strategy improves the length generalization of the doc2doc models. Compared to the MR strategy, our proposed DLS effectively balances the amount of the text of different lengths. The experimental results shows that our proposed method is capable of handling the text with arbitrary length.

### 6.2 The Length Bias Problem

Although the length bias problem has not been explore in the field of DNMT, there still exists several studies emphasis on the length extrapolation problem. Press et al. (2021) proposed to solve the length extrapolation problem by introducing local assumption as the inductive bias in the positional encoding. Following this work, Chi et al. (2022) and Sun et al. (2022a) further proposed new positional encoding methods to overcome this issue. Additionally, (Ruoss et al., 2023) introduced random positional encoding training strategy to overcome the length extrapolation problem, and achieved remarkable progress within the field of language modeling.

## 7 Conclusion

In this work, we aim to address the issue of length bias in the training of the doc2doc DNMT model. To achieve this objective, we propose several methods, including dynamic length sampling, length aware attention and sliding decoding. We conduct experiments on multiple publicly available datasets,

and the results demonstrate a significant improvement achieved by our method. Further analysis indicates that our approach can enhance the length generalization capability of the model effectively.

## 8 Acknowledgement

We would like to express our gratitude to the ICT computing platform and the technical service team for providing GPU resources.

Furthermore, we thank the anonymous reviewers for their thorough review and valuable feedback.

## Limitations

Although our proposed methods significantly improve the translation quality and the length generalization capability, there still exist some limitations: (1) the slide decoding can not further improve the translation quality as the error accumulation problem of auto-regression model has not been solved; (2) the decoding consumption using SD is slightly higher than the "segment then decoding" method.

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

## A  The Proof of the Length Aware Attention

The entropy of the attention mechanism can be calculated using the following formula[4]:

$$\mathcal{H}_i = -\sum_{j=1}^{n} a_{i,j} \log a_{i,j}$$

$$= \log \sum_{j=1}^{n} e^{\lambda \boldsymbol{q}_i \cdot \boldsymbol{k}_j} - \frac{\sum_{j=1}^{n} e^{\lambda \boldsymbol{q}_i \cdot \boldsymbol{k}_j} (\lambda \boldsymbol{q}_i \cdot \boldsymbol{k}_j)}{\sum_{j=1}^{n} e^{\lambda \boldsymbol{q}_i \cdot \boldsymbol{k}_j}},$$

$$(6)$$

where $\mathcal{H}_i$ indicate the attention entropy of the $i$-th token, $\lambda$ is the scale factor, and $a_{i,j}$ is the attention weight. Let $\boldsymbol{s}_{i,j} = \boldsymbol{q}_i \cdot \boldsymbol{k}_j$ and $\boldsymbol{p}_{i,j} = \frac{\boldsymbol{s}_{i,j}}{\sum_{j=1}^{n} e^{\lambda \boldsymbol{s}_{i,j}}}$, we get:

$$\mathcal{H}_i = \log \sum_{j=1}^{n} e^{\lambda \boldsymbol{s}_{i,j}} - \lambda \sum_{j=1}^{n} \boldsymbol{p}_{i,j} \boldsymbol{s}_{i,j},$$

$$= \log n + \log \frac{1}{n} \sum_{j=1}^{n} e^{\lambda \boldsymbol{s}_{i,j}} - \lambda \sum_{j=1}^{n} \boldsymbol{p}_{i,j} \boldsymbol{s}_{i,j}$$

$$(7)$$

According to Mean-field theory, we could change the order of computation between the exponential function and summation:

$$\mathcal{H}_i \approx \log n + \lambda \bar{\boldsymbol{s}}_i - \lambda \sum_{j=1}^{n} \boldsymbol{p}_{i,j} \boldsymbol{s}_{i,j} \qquad (8)$$

where $\bar{\boldsymbol{s}}_i = \sum_{j=1}^{n} \boldsymbol{s}_{i,j}/n$. Considering the properties of the softmax function, we can obtain further approximations:

$$\mathcal{H}_i \approx \log n + \lambda(\bar{\boldsymbol{s}}_i - \lambda \boldsymbol{s}_{max}) \qquad (9)$$

Thus, we get:

$$\lambda \propto \log n, \qquad (10)$$

where $\lambda$ is proportional to $\log n$.

---

[4]Our derivation is motivated by the following blog: https://spaces.ac.cn/archives/9034