# OpenReview forum: "Addressing the Length Bias Challenge in Document-Level Neural Machine Translation"
_EMNLP/2023/Conference — EMNLP 2023 Findings_

### Official Review · Reviewer_ap6j · 2023-08-05

**Soundness:** 3

**Excitement:**

3: Ambivalent: It has merits (e.g., it reports state-of-the-art results, the idea is nice), but there are key weaknesses (e.g., it describes incremental work), and it can significantly benefit from another round of revision. However, I won't object to accepting it if my co-reviewers champion it.

**Paper Topic And Main Contributions:**

This paper addresses the length bias problem in document-level translation.
Conventional methods are effective only for long or short sequences and cannot handle both sequences.
The authors proposed a method to extend the length generation capability to handle both long and short sequences.
They proposed the following three methods.
1. sampling to learn sequences of various lengths during learning
2. scaling in Attention by the average length of the series in the training epoch
3. sliding decoding

**Questions For The Authors:**

A. What is the distribution of series lengths in the data set? Did the similar distributions of these three datasets improve the accuracy across all of them?
B. How many are the maximum series lengths of each?

**Reasons To Accept:**

The paper proposes a combination of three methods. Although each method is simple, they all seem effective for the length bias problem.
Comparison experiments have been conducted on many datasets and comparison systems.
The experimental results suggest that the proposed methods are effective.
The analysis is well done and shows that the proposed method is effective for both long and short series.

**Reasons To Reject:**

The authors describe a problem when decoding sentences that are shorter or longer than the maximum length being trained, but how often this problem occurs should be described.
There is no data on the distribution of series lengths in the dataset.
The dataset contains a variety of series lengths, and the score of the proposed method may vary greatly depending on the distribution of these series lengths.

**Reproducibility:**

4: Could mostly reproduce the results, but there may be some variation because of sample variance or minor variations in their interpretation of the protocol or method.

**Reviewer Confidence:**

3: Pretty sure, but there's a chance I missed something. Although I have a good feel for this area in general, I did not carefully check the paper's details, e.g., the math, experimental design, or novelty.

**Typos Grammar Style And Presentation Improvements:**

In Section 2, 'The Transformer' and 'The doc2doc DNMT model' are proper nouns, so we do not need 'The'.

---

> ### Author Rebuttal · Authors · 2023-08-29
>
> We appreciate your careful consideration of our paper and would like to take this opportunity to respond to the concerns you have raised.
>
> ## Q1: How often does the length-bias problem occur?
> Thank you for your attention to the aspect of the length-bias problem. This problem is indeed prevalent in real-world scenarios. As depicted in Figure 6, training datasets often exhibit a concentrated distribution of lengths. However, during the decoding phase, users do not limit the translation system to translating only texts of the same length as those in the training set. Instead, users also apply translation systems to sentence-level translation or even longer document-level translation, addressing scenarios that traditional DNMT methods struggle to handle. Hence, we believe this problem presents a valuable opportunity for practical application.
>
>
> ## Q2: What is the length distribution of the dataset?
> Thanks again for your advice. We will include the original length distribution of these datasets in our final paper. In addition, since these datasets are usually collected from similar sources, their length distributions are concentrated. However, as our method sampling the length of the dataset, the model trained with DLS, LAA and SD can be applied to translation tasks of any length.
>
> ## Q3: Typos Grammar Style and Presentation Improvements.
> Thank you for your careful examination of our paper. We will try our best to further improve the writing of this paper.
>
> ## Conclustion
> We sincerely appreciate your thoughtful feedback and constructive criticisms of our work. We have taken a thorough review of your comments and have taken the time to address each of the points you raised in detail. If our responses align with your expectations, we would greatly appreciate an increase in our score. If you have any concerns or doubts about our response, please feel free to contact us during the discussion period.

---

### Official Review · Reviewer_VHhW · 2023-08-05

**Soundness:** 3

**Excitement:**

4: Strong: This paper deepens the understanding of some phenomenon or lowers the barriers to an existing research direction.

**Paper Topic And Main Contributions:**

In this paper, the authors focus on the length bias problem in Document-level NMT, which results in significant translation quality degradation in terms of sentence length.
To address this issue, the authors optimize the Doc-NMT model in three levels, i.e., training data sampling, length-aware attention, and sliding windows decoding strategy.
As shown in their experimental results, the proposed method could achieve the best score on three widely-used datasets and significantly mitigate the length bias problem in the document-level translation task.


**Reasons To Accept:**

This paper proposes a simple yet effective method to mitigate the length-bias problem in Doc-NMT, which achieves the sota score in three datasets and significantly mitigates the length-bias problem in this task.


**Reasons To Reject:**

Although the whole method is effective, the sliding decoding strategy seems cannot be effective as expected, while lacking the declaration of the window-size. And no further corresponding exploration for this failure in Sec.5.2.
Besides, the performance of other relative methods on this length-bias problem is also important to be analyzed and compared.
Lack the statistical analysis for the variation of the sentence length after sampling.


**Reproducibility:**

4: Could mostly reproduce the results, but there may be some variation because of sample variance or minor variations in their interpretation of the protocol or method.

**Reviewer Confidence:**

4: Quite sure. I tried to check the important points carefully. It's unlikely, though conceivable, that I missed something that should affect my ratings.

**Typos Grammar Style And Presentation Improvements:**

Could you exhibit the improvement of your method in the Abstract and Introduction?

---

> ### Author Rebuttal · Authors · 2023-08-29
>
> Your feedback is greatly appreciated, and we would like to address your main points of concern to provide a clearer understanding of our work.
>
> ## Q1: Why the proposed Slide Decoding can not improve the BLEU further?
> We propose the **Sliding Decoding (SD)** approach in response to the potential significant incoherence introduced by truncating the entire documents into multiple segments. Our proposed method offers two notable advantages:
> 1. Our approach can further **enhance the length generalization performance of the DNMT model**. As shown in Figure 5 in our paper, the SD method significantly improves the BLEU score when the sequence length exceeds the maximum length used during training. In order to ensure a fair comparison, we split all the test sets that were not decoded using the SD method into segments with a maximum length of 512 in the main experiment. As a result, **the superiority of our approach was not fully demonstrated in the main experiments**.
> 2. Due to its ability to leverage as much contextual information as possible, our proposed Slide Decoding can further **enhance the fluency of translations**. However, this aspect may not be well-reflected by the BLEU metric. As a result, the advantage of our method was not fully showcased in the experiments.
>
> Furthermore, the **error accumulation problem** may be another reason for the limited improvement in BLEU scores observed in the SD method. We will explore and address this problem in our future work.
>
> ## Q2: What is the window-size of Slide Decoding in the experiments?
> Thank you for meticulously examining our paper. As our motivation is to maximize the contextual information, the windows-size of Slide Decoding is always **aligned with the maximum sequence length during training**, which is equal to 512.
>
> ## Q3: How do other related methods perform on this length-bias problem?
> We appreciate your attention to the length-bias problem. In response to your comment, we would like to address this concern and provide more information regarding our work.
> 1. As far as our knowledge extends, **we are the first highlighting the length bias problem in the domain of document-level machine translation**. Our work emphasizes the significance of addressing this problem and has resulted in significant improvements.
> 2. In order to provide a comprehensive evaluation and draw comparisons with length generalization efforts in other domains, we have conducted a thorough comparison with the well-recognized ALIBI method [1], known for its exceptional length extrapolation capabilities.
>
> | Model vs dataset       | TED    |        |        |        | Europarl |        |        |        | News   |        |        |        |
> | ---------------------- | ------ | ------ | ------ | ------ | -------- | ------ | ------ | ------ | ------ | ------ | ------ | ------ |
> |                        | s-BLEU | d-BLEU | s-chrF | d-chrF | s-BLEU   | d-BLEU | s-chrF | d-chrF | s-BLEU | d-BLEU | s-chrF | d-chrF |
> | ALIBI-doc              | 19.65  | 26.30  | 47.22  | 68.26  | 29.99    | 32.54  | 60.19  | 69.28  | 12.67  | 22.83  | 36.19  | 59.91  |
> | ALIBI-FT               | 20.85  | 27.55  | 49.54  | 69.71  | 29.64    | 32.55  | 59.76  | 69.41  | 17.37  | 24.02  | 43.49  | 60.79  |
> | Transformer-FT         | 24.31  | 28.48  | 54.70  | 70.51  | 31.16    | 33.58  | 61.29  | 69.91  | 23.96  | 27.33  | 53.27  | 64.98  |
> | Transformer+DLS+LAA    | 24.93  | 28.95  | 55.18  | 70.69  | 31.85    | 34.32  | 61.26  | 70.01  | 24.57  | 28.52  | 53.13  | 65.47  |
> | Transformer+DLS+LAA+SD | 24.60  | 28.43  | 55.16  | 70.55  | 31.63    | 34.33  | 60.91  | 69.93  | 23.72  | 27.95  | 52.50  | 65.20  |
> | G-Trans-FT             | 25.07  | 28.86  | 55.65  | 70.79  | 32.38    | 34.51  | 62.15  | 70.32  | 25.87  | 27.82  | 55.71  | 65.14  |
> | G-Trans+DLS+LAA        | 25.37  | 29.07  | 55.76  | 70.81  | 32.67    | 34.81  | 62.06  | 70.23  | 26.70  | 28.66  | 55.96  | 65.24  |
> | G-Trans+DLS+LAA+SD     | 24.87  | 28.53  | 55.34  | 70.51  | 32.67    | 34.82  | 61.99  | 70.17  | 26.56  | 28.52  | 55.78  | 65.09  |
>
>
> | Model vs sequence length | 32    | 64    | 128   | 256   | 512   | 1024  |
> | ------------------------ | ----- | ----- | ----- | ----- | ----- | ----- |
> | Transformer-FT           | 29.75 | 31.00 | 32.55 | 33.04 | 33.58 | 28.28 |
> | ALIBI-FT                 | 13.46 | 18.90 | 26.87 | 30.66 | 32.55 | 30.54 |
> | Transformer+Our Method   | 33.48 | 33.73 | 33.90 | 34.11 | 34.32 | 34.33 |
>
>
> As the shown in the first above table, **our proposed method outperforms ALIBI on all datasets**, indicating that our method is better suited for DNMT. Additionally, we compared the length generalization of the ALIBI method. The experimental results within the second table demonstrated that ALIBI exhibits better length extrapolation than the baseline method when decoding sequences exceed the training length, but there still exists a gap between ALIBI and our method. We attribute this difference to the fact that ALIBI's generalization performance relies on local assumptions within the attention mechanism, which contradict the essence of DNMT that utilize contexutal information wherever possible. Furthermore, in the context of encoder-decoder architecture for machine translation, the cross-attention mechanism cannot benefit from this assumption. These two reasons combined contribute to ALIBI's inability to surpass our proposed method in the context of discourse translation.
>
> [1] Press et al., 2021, Train short, test long: Attention with linear biases enables input length extrapolation.
>
>
> ## Q4: Lack the statistical analysis for the variation of the sentence length after sampling.
> Firstly, it's important to note that **our sampling method does not directly alter the length of individual sentences**. Instead, it **dynamically adjusts the number of sentences to modify the length of text segments**. This approach was chosen to ensure that the inherent structure and coherence of sentences are preserved while achieving our intended text length variations.
>
> To better illustrate the impact of our sampling method on sentence length distribution, we have included **Figure 6** in our paper. This figure demonstrates the distribution of segment lengths before and after the application of our sampling method. As depicted in the figure, after implementing our sampling technique, the distribution of segment lengths becomes more uniform. This observation supports the effectiveness of our approach in achieving the desired outcome without compromising the coherence of individual sentences.
>
>
> ## Q5: The abstract and the introduction could be further improved.
> Thanks for your constructive advise. We will try our best to further improve the writing of this paper.
>
> ## Conclusion
> We sincerely appreciate your thoughtful feedback and constructive criticisms on our work. We have taken a thorough review of your comments and have taken the time to address each of the points you raised in detail. If our responses align with your expectations, we would greatly appreciate an increase in our score. If you have any concerns or doubts about our response, please feel free to contact us during discussion period.

---

### Official Review · Reviewer_Wr8v · 2023-08-08

**Soundness:** 4

**Excitement:**

3: Ambivalent: It has merits (e.g., it reports state-of-the-art results, the idea is nice), but there are key weaknesses (e.g., it describes incremental work), and it can significantly benefit from another round of revision. However, I won't object to accepting it if my co-reviewers champion it.

**Paper Topic And Main Contributions:**

This paper proposes methods to improve document-level neural machine translation (DNMT) by addressing the length bias problem, where model performance degrades on sentences much shorter/longer than the maximum length seen during training. Experiments across 3 widely-used datasets demonstrate consistent and sizable improvements in BLEU and chrF over competitive baseline DNMT methods. The techniques are also shown to improve length generalization.


**Main contributions:**
1. Dynamic length sampling during training to expose the model to more varied sentence lengths.
2. Length-aware attention to help focus on relevant target words for long sentences.
3. Sliding window decoding to handle test sentences exceeding the max length.

**Reasons To Accept:**

1. The problem being addressed is relevant and the ideas are intuitively motivated to address the length bias problem in document translation.
2. The methods seem simple to implement on top of standard DNMT models.
3. The paper is clearly written and easy to follow.

**Reasons To Reject:**

1. The sliding window decoding actually seems to hurt performance slightly even though it should help length generalization. Error accumulation may be an issue.
2. More analysis could be done to understand the effects of the different components, e.g. how attention patterns change with length-aware scaling.
3. The techniques are evaluated on standard datasets. Testing on challenging length generalization datasets would better showcase their benefits.
4. There is no comparison to other recent methods (if any) that improve length generalization for sequence models.

---
post-rebuttal: most concerns have been addressed.

**Reproducibility:**

3: Could reproduce the results with some difficulty. The settings of parameters are underspecified or subjectively determined; the training/evaluation data are not widely available.

**Reviewer Confidence:**

3: Pretty sure, but there's a chance I missed something. Although I have a good feel for this area in general, I did not carefully check the paper's details, e.g., the math, experimental design, or novelty.

---

> ### Author Rebuttal · Authors · 2023-08-29
>
> Your feedback is greatly appreciated, and we would like to address your main points of concern to provide a clearer understanding of our work.
>
> ## Q1: Why the proposed Slide Decoding can not improve the BLEU further?
> We propose the **Sliding Decoding (SD)** approach in response to the potential significant incoherence introduced by truncating the entire documents into multiple segments. Our proposed method offers two notable advantages:
> 1. Our approach can further **enhance the length generalization performance of the DNMT model**. As shown in Figure 5 in our paper, the SD method significantly improves the BLEU score when the sequence length exceeds the maximum length used during training. In order to ensure a fair comparison, we split all the test sets that were not decoded using the SD method into segments with a maximum length of 512 in the main experiment. As a result, **the superiority of our approach was not fully demonstrated in the main experiments**.
> 2. Due to its ability to leverage as much contextual information as possible, our proposed Slide Decoding can further **enhance the fluency of translations**. However, this aspect may not be well-reflected by the BLEU metric. As a result, the advantage of our method was not fully showcased in the experiments.
>
> Furthermore, the **error accumulation problem** may be another reason for the limited improvement in BLEU scores observed in the SD method. We will explore and address this problem in our future work.
>
> ## Q2: Lack of analytical experiments on Length Aware Attention.
> Thank you for your constructive advise, and we have incorporated additional experiments on Length Aware Attention as follows. To verify the effectiveness of our proposed **Length Aware Attention (LAA)**, we conducted a statistical analysis of the average entropy of the attention mechanism when the model translates sentences and documents of length 512. As shown in the table below, it can be observed that the entropy of the attention mechanism is more stable after applying the LAA method, which  indicates that after applying the LAA mechanism, the model demonstrates better consistency in handling sentence-level and document-level text. Furthermore, by applying the DLS and LAA method, the entropy of attention when translating the document is lower than that of the FT method, indicating that the model concentrates more on the long-range contexts.
>
> | Entropy of Model    | sentence | document | Δ    |
> | ------------------- | -------- | -------- | ---- |
> | Transformer-FT      | 2.65     | 4.0      | 1.35 |
> | Transformer-DLS     | 2.51     | 3.95     | 1.44 |
> | Transformer-DLS-LAA | 2.74     | 3.94     | 1.20 |
>
> ## Q3: Testing on challenging length generalization datasets would better showcase the benefits.
> Thanks again for your advise, following [1], we collect ContraPro as **discourse dense dataset** to measure our method. To further increase the challenging of the dataset for length generalization, we truncate the dataset into different length. We evaluate the translation quality using d-BLEU, and the results are shown in the following table.
>
> | Model on discourse dense dataset | d-BLEU |
> | -------------------------------- | ------ |
> | Transformer-FT                   | 19.01  |
> | Transformer-DLS                  | 19.57  |
> | Transformer-DLS-LAA              | 19.90  |
>
> As shown in the above table, our proposed method performs significantly better than the baseline method, indicating the effectiveness of our method in the context of discourse translation.
>
>
> [1] Post et al., 2023, Escaping the sentence-level paradigm in machine translation.
>
>
> ## Q4: There is no comparison to other recent methods that improve length generalization for sequence models.
> We appreciate your attention to the aspect of length generalization in sequence models. In response to your comment, we would like to address this concern and provide more information regarding our work.
> 1. As far as our knowledge extends, **we are the first highlighting the length bias problem in the domain of document-level machine translation**. Our work emphasizes the significance of addressing this problem and has resulted in significant improvements.
> 2. In order to provide a comprehensive evaluation and draw comparisons with length generalization efforts in other domains, we have conducted a thorough comparison with the well-recognized ALIBI method [2], known for its exceptional length extrapolation capabilities.
>
> | Model vs dataset       | TED    |        |        |        | Europarl |        |        |        | News   |        |        |        |
> | ---------------------- | ------ | ------ | ------ | ------ | -------- | ------ | ------ | ------ | ------ | ------ | ------ | ------ |
> |                        | s-BLEU | d-BLEU | s-chrF | d-chrF | s-BLEU   | d-BLEU | s-chrF | d-chrF | s-BLEU | d-BLEU | s-chrF | d-chrF |
> | ALIBI-doc              | 19.65  | 26.30  | 47.22  | 68.26  | 29.99    | 32.54  | 60.19  | 69.28  | 12.67  | 22.83  | 36.19  | 59.91  |
> | ALIBI-FT               | 20.85  | 27.55  | 49.54  | 69.71  | 29.64    | 32.55  | 59.76  | 69.41  | 17.37  | 24.02  | 43.49  | 60.79  |
> | Transformer-FT         | 24.31  | 28.48  | 54.70  | 70.51  | 31.16    | 33.58  | 61.29  | 69.91  | 23.96  | 27.33  | 53.27  | 64.98  |
> | Transformer+DLS+LAA    | 24.93  | 28.95  | 55.18  | 70.69  | 31.85    | 34.32  | 61.26  | 70.01  | 24.57  | 28.52  | 53.13  | 65.47  |
> | Transformer+DLS+LAA+SD | 24.60  | 28.43  | 55.16  | 70.55  | 31.63    | 34.33  | 60.91  | 69.93  | 23.72  | 27.95  | 52.50  | 65.20  |
> | G-Trans-FT             | 25.07  | 28.86  | 55.65  | 70.79  | 32.38    | 34.51  | 62.15  | 70.32  | 25.87  | 27.82  | 55.71  | 65.14  |
> | G-Trans+DLS+LAA        | 25.37  | 29.07  | 55.76  | 70.81  | 32.67    | 34.81  | 62.06  | 70.23  | 26.70  | 28.66  | 55.96  | 65.24  |
> | G-Trans+DLS+LAA+SD     | 24.87  | 28.53  | 55.34  | 70.51  | 32.67    | 34.82  | 61.99  | 70.17  | 26.56  | 28.52  | 55.78  | 65.09  |
>
>
> | Model vs sequence length | 32    | 64    | 128   | 256   | 512   | 1024  |
> | ------------------------ | ----- | ----- | ----- | ----- | ----- | ----- |
> | Transformer-FT           | 29.75 | 31.00 | 32.55 | 33.04 | 33.58 | 28.28 |
> | ALIBI-FT                 | 13.46 | 18.90 | 26.87 | 30.66 | 32.55 | 30.54 |
> | Transformer+Our Method   | 33.48 | 33.73 | 33.90 | 34.11 | 34.32 | 34.33 |
>
>
> As the shown in the first above table, our proposed method outperforms ALIBI on all datasets, indicating that **our method is better suited for DNMT**. Additionally, we compared the length generalization of the ALIBI method. The experimental results within the second table demonstrated that ALIBI exhibits better length extrapolation than the baseline method when decoding sequences exceed the training length, but there still exists a gap between ALIBI and our method. We attribute this difference to the fact that ALIBI's generalization performance relies on local assumptions within the attention mechanism, which contradict the essence of DNMT that utilize contexutal information wherever possible. Furthermore, in the context of encoder-decoder architecture for machine translation, the cross-attention mechanism cannot benefit from this assumption. These two reasons combined contribute to ALIBI's inability to surpass our proposed method in the context of discourse translation.
>
> [2] Press et al., 2021, Train short, test long: Attention with linear biases enables input length extrapolation.
>
> ## Conclusion
> We sincerely appreciate your thoughtful feedback and constructive criticisms on our work. We have taken a thorough review of your comments and have taken the time to address each of the points you raised in detail. If our responses align with your expectations, we would greatly appreciate an increase in our score. If you have any concerns or doubts about our response, please feel free to contact us during discussion period.

---

### Official Review · Reviewer_J2RQ · 2023-08-11

**Typos Grammar Style And Presentation Improvements:** N/A
**Soundness:** 3

**Excitement:**

3: Ambivalent: It has merits (e.g., it reports state-of-the-art results, the idea is nice), but there are key weaknesses (e.g., it describes incremental work), and it can significantly benefit from another round of revision. However, I won't object to accepting it if my co-reviewers champion it.

**Missing References:**

N/A

**Paper Topic And Main Contributions:**

This paper addresses document-level neural machine translation (NMT).

The authors introduce the "length bias" problem, where the translation quality degrades significantly as the decoding length deviates from the maximum sentence length.
To mitigate this bias, they propose a novel document-level NMT training method: 1) Dynamic length sampling (DLS) 2) Length-aware attention (LAA) 3) Sliding decoding (SD).

When integrating their proposed method into G-transformer, they show significant improvements on all datasets.

**Questions For The Authors:**

- A significant amount of previous research ([1], [2]) adopts two-stage training strategy: pre-training at the sentence-level and fine-tuning at the document-level. Therefore, it would be recommended to empirically show if the proposed method is also useful for fine-tuning a pre-trained sentence-level NMT model at the document-level (e.g., Trans-FT + DLS + LAA, G-Trans-FT + LAA). The dynamic length sampling based on curriculum learning (in Figure 2) is questionable.

[2] Voita et al., 2019, When a Good Translation is Wrong in Context: Context-Aware Machine Translation Improves on Deixis, Ellipsis, and Lexical Cohesion

[3] Maruf et al., 2019, Selective Attention for Context-aware Neural Machine Translation

**Reasons To Accept:**

1. The motivation of this paper is clear.

2. The three specific training methods are novel.

**Reasons To Reject:**

Most importantly, the problems that the authors claim are needed to be well-supported by either logic or experiments.

1. The authors claim that LAA is motivated by the assumption that the document-level NMT model may interfere with the attention mechanism since it needs to handle a wide range of context. However, [1] explains that the actual amount of context needed is not as much as the previous few (1~3) sentences. It might be related to the issue of entropy divergence, which is newly addressed by the authors. The paper is stated that it would explain in an appendix but it doesn't exist.

2. The authors also claim that SD is motivated by the assumption that the contextual information may be lost when decoding longer sequences in segments. Considering [1], it is not expected to be a problem, as the necessary context is preserved even if it is split.
The lack of performance gains also supports this suspicion.

[1] Fernandes et al., 2021, Measuring and Increasing Context Usage in Context-Aware Machine Translation

**Reproducibility:**

3: Could reproduce the results with some difficulty. The settings of parameters are underspecified or subjectively determined; the training/evaluation data are not widely available.

**Reviewer Confidence:**

3: Pretty sure, but there's a chance I missed something. Although I have a good feel for this area in general, I did not carefully check the paper's details, e.g., the math, experimental design, or novelty.

---

> ### Author Rebuttal · Authors · 2023-08-29
>
> Thanks for your thorough review and constructive feedback. We are glad that you found our approach to be innovative. In response to the concern raised about our paper, we summarize our response as follows.
>
> ## Q1: The proof of entropy divergense problem in attention mechanism is missing.
> Thank you very much for pointing out the absence of our appendix section. We apologize for omitting this part during the paper submission, and we would like to assure you that we are taking immediate steps to rectify this oversight in our final paper.
>
> In addition, to verify the effectiveness of our proposed **Length Aware Attention (LAA)**, we conducted a statistical analysis of the average entropy of the attention mechanism when the model translates sentences and documents of length 512. As shown in the table below, it can be observed that the entropy of the attention mechanism is more stable after applying the LAA method, which indicates that after applying the LAA mechanism, the model demonstrates better consistency in handling sentence-level and document-level text. Furthermore, by applying the DLS and LAA method, the entropy of attention when translating the document is lower than that of the FT method, indicating that the model concentrates more on the long-range contexts. To further demonstrate the effect of Length Aware Attention, we also plan to add the Attention visualization results in the appendix of our paper.
>
> | Model               | sentence | document | Δ    |
> | ------------------- | -------- | -------- | ---- |
> | Transformer-FT      | 2.65     | 4.0      | 1.35 |
> | Transformer-DLS     | 2.51     | 3.95     | 1.44 |
> | Transformer-DLS-LAA | 2.74     | 3.94     | 1.20 |
>
> ## Q2: Is the use of longer contexts necessary?
> Although many previous studies have shown that the most informative contexts tend to be concentrated within the previous one to three sentences, we contend that this observation may be attributed to the limitations of conventional sent2sent DNMT model in effectively harnessing the entirety of contextual information. Recently, numerous doc2doc DNMT studies have demonstrated its superior capability in leveraging contextual information compared to the sentence-to-sentence approaches [1,2]. In addition, as shown in the following table, our experiment reveals that **even vanilla Transformer can benefit from an increased amount of contextual information** when there is an ample amount of training data.
>
> | Maximum Seq | Europarl7 |        |        |        |
> | ----------- | --------- | ------ | ------ | ------ |
> |             | s-BLEU    | d-BLEU | s-chrF | d-chrF |
> | sent        | 30.33     | 32.45  | 59.99  | 68.11  |
> | 64          | 30.56     | 32.77  | 60.13  | 68.32  |
> | 128         | 30.97     | 33.13  | 60.40  | 68.47  |
> | 512         | 31.16     | 33.58  | 61.29  | 69.91  |
>
>
> [1] Bao et al., 2021, G-transformer for document-level machine translation.
> [2] Liu et al., 2020, Multilingual denoising pre-training for neural machine translation.
>
> ## Q3: Why the proposed Slide Decoding can not improve the BLEU further?
> We propose the **Sliding Decoding (SD)** approach in response to the potential significant incoherence introduced by truncating the entire documents into multiple segments. Our proposed method offers two notable advantages:
> 1. Our approach can further **enhance the length generalization performance of the DNMT model**. As shown in Figure 5 in our paper, the SD method significantly improves the BLEU score when the sequence length exceeds the maximum length used during training. In order to ensure a fair comparison, we split all the test sets that were not decoded using the SD method into segments with a maximum length of 512 in the main experiment. As a result, **the superiority of our approach was not fully demonstrated in the main experiments**.
> 2. Due to its ability to leverage as much contextual information as possible, our proposed Slide Decoding can further **enhance the fluency of translations**. However, this aspect may not be well-reflected by the BLEU metric. As a result, the advantage of our method was not fully showcased in the experiments.
>
> Furthermore, the **error accumulation problem** may be another reason for the limited improvement in BLEU scores observed in the SD method. We will explore and address this problem in our future work.
>
> ## Q4: Whether the proposed methods are useful for fine-tuning a pre-trained sentence-level NMT model at the document-level?
> The answer to this question is yes. We have conducted series experiments to reveal the effectiveness of our proposed method. As shown in Table 1 in the paper, **our approach significantly outperforms the 2-stage baseline methods** (Transformer-FT, G-Trans-FT). In addition, in Table 2 of the ablation study, both System 4 and System 5 utilize the 2-stage approach, and when compared against System 2 and System 3 which employ the DLS method, they collectively demonstrate the effectiveness of our approach.
>
> To further demonstrate the effectiveness of our proposed dynamic length sampling method, we simply apply length sampling training stage without dynamic adjusting the probability on a pre-trained sentence-level Transformer model. However, due to time constraints, **we will give our experimental results in the discussion phase**.
>
> ## Conclusion
> We sincerely appreciate your thoughtful feedback and constructive criticisms on our work. We have taken a thorough review of your comments and have taken the time to address each of the points you raised in detail. If our responses align with your expectations, we would greatly appreciate an increase in our score. If you have any concerns or doubts about our response, please feel free to contact us during discussion period.

---

### Meta-Review · Area_Chair_ZCPN · 2023-09-18

**Recommendation:** 4

**Metareview:**

This paper proposes and addresses a novel problem - the "length bias" problem - that of document level machine translation models struggling to translate documents which are much smaller or larger than those in the training data.
Using a number of strategies during training (sampling and length-normalised attention) and decoding (sliding decoding) they show strong results on several open datasets and mitigate the length bias problem in the document-level translation task.
One reviewer noted that it would have been better to test their model on datasets which display the "length bias" problem instead of standard datasets and the authors provided a table with some relevant results. The authors also provided a comparison with other recent document level MT methods and their methods perform better.
There were four reviews and lengthy rebuttals with new results and clarifications which led to one reviewer increasing their score.

---

### Decision · Program_Chairs · 2023-10-07

**Decision:**

Accept-Findings

**Comment:**

This paper proposes and addresses a novel problem - the "length bias" problem - that of document level machine translation models struggling to translate documents which are much smaller or larger than those in the training data.
Using a number of strategies during training (sampling and length-normalised attention) and decoding (sliding decoding) they show strong results on several open datasets and mitigate the length bias problem in the document-level translation task.
One reviewer noted that it would have been better to test their model on datasets which display the "length bias" problem instead of standard datasets and the authors provided a table with some relevant results. The authors also provided a comparison with other recent document level MT methods and their methods perform better.
There were four reviews and lengthy rebuttals with new results and clarifications which led to one reviewer increasing their score.